# Dysregulation of Rho GTPases in Human Cancers

**DOI:** 10.3390/cancers12051179

**Published:** 2020-05-07

**Authors:** Haiyoung Jung, Suk Ran Yoon, Jeewon Lim, Hee Jun Cho, Hee Gu Lee

**Affiliations:** 1Immunotherapy Research Center, Korea Research Institute of Bioscience and Biotechnology, Daejeon 34141, Korea; haiyoung@kribb.re.kr (H.J.); sryoon@kribb.re.kr (S.R.Y.); ljw8796@kribb.re.kr (J.L.); 2Department of Biomolecular Science, KRIBB School of Bioscience, Korea University of Science and Technology (UST), Yuseong-gu, Daejeon 34141, Korea

**Keywords:** Rho GTPases, cancer, migration, metastasis, chemoresistance

## Abstract

Rho GTPases play central roles in numerous cellular processes, including cell motility, cell polarity, and cell cycle progression, by regulating actin cytoskeletal dynamics and cell adhesion. Dysregulation of Rho GTPase signaling is observed in a broad range of human cancers, and is associated with cancer development and malignant phenotypes, including metastasis and chemoresistance. Rho GTPase activity is precisely controlled by guanine nucleotide exchange factors, GTPase-activating proteins, and guanine nucleotide dissociation inhibitors. Recent evidence demonstrates that it is also regulated by post-translational modifications, such as phosphorylation, ubiquitination, and sumoylation. Here, we review the current knowledge on the role of Rho GTPases, and the precise mechanisms controlling their activity in the regulation of cancer progression. In addition, we discuss targeting strategies for the development of new drugs to improve cancer therapy.

## 1. Introduction

Rho GTPases belong to the Ras superfamily, and comprise more than 20 members that have been classified into eight subfamilies (Rho, Rac, Cdc42, RhoD/RhoF, RhoH, RhoU/RhoV, Rnd, and RhoBTB) on the basis of their structure and function [1]. Like other Ras superfamily members, most Rho GTPases cycle between an inactive guanosine diphosphate (GDP)-bound state in the cytoplasm, and an active guanosine triphosphate (GTP)-bound state in the cell membrane. This cycling is precisely regulated by three classes of proteins: Rho guanine nucleotide exchange factors (RhoGEFs), Rho GTPase-activating proteins (RhoGAPs), and Rho guanine nucleotide dissociation inhibitors (RhoGDIs). Rho GTPases are activated by RhoGEFs, which catalyze the exchange of GDP for GTP, and they are inactivated by RhoGAPs, which promote the intrinsically slow GTP-hydrolysis activity of Rho GTPases [2,3]. RhoGDIs bind to GDP-bound Rho GTPases and regulate their spatiotemporal activity. In addition, their activity is regulated by post-translational modifications (PTMs), including lipid modification, phosphorylation, ubiquitination, and sumoylation, which lead to changes in cell behavior [4].

Activated Rho GTPases interact with various effector proteins, and transduce downstream signaling that regulates numerous cellular functions, including cell motility, cell polarity, and cell proliferation [5]. In various diseases, including cancer, Rho GTPase activity is abnormally regulated through altered expression and PTMs, as well as by their regulator and effector proteins [1]. Recent studies have identified activating mutations in Rho GTPases, such as RhoA, Rac1, and Cdc42, in various human cancers [6]. Accumulating evidence demonstrates that the dysregulation of Rho GTPase signaling is closely associated with tumorigenesis and malignant phenotypes, including transformation, cell cycle progression, migration, invasion, metastasis, and drug resistance [7]. Therefore, Rho GTPases and their regulator proteins are considered attractive targets for therapeutic intervention. Here, we review recent findings on the role of Rho GTPases in cancer progression and the precise mechanisms controlling their activity. In addition, we discuss potential opportunities for therapeutic intervention.

## 2. Rho GTPases and Their Direct Regulator Partners

Rho GTPases regulate a broad range of biological processes, including actin cytoskeletal rearrangements, cell motility, proliferation, differentiation, senescence, vesicle trafficking, and cell survival. Growth factor receptors, integrins, E-cadherin, and chemokine receptors promote the activation of Rho GTPases in response to extracellular stimuli [1]. Rho GTPases are classified as classical or atypical (Figure 1). Classical Rho GTPases, such as RhoA, Rac1, and Cdc42, act as molecular switches, cycling between an inactive GDP-bound form in the cytoplasm, and an active GTP-bound form in the plasma membrane [8]. This cycling is regulated by the opposing actions of RhoGEFs and RhoGAPs. Rho GTPases are activated by RhoGEFs, which promote GDP dissociation, an intrinsically slow process, thus stimulating the exchange for GTP that is present at higher concentrations in the cytosol. RhoGEFs are classified into two families: the Dbl-homology domain family and the Dock homology region domain family. These domains catalyze the GDP/GTP nucleotide exchange [9]. On the other hand, RhoGAPs accelerate the intrinsic GTP-hydrolysis activity of Rho GTPases, thus returning them to an inactive state. RhoGAPs have a highly conserved GAP domain, which binds to Rho GTPases to promote their GTPase activity [3]. Thus, RhoGEFs and RhoGAPs are considered to be positive and negative regulators of Rho GTPases, respectively.

In addition to RhoGEFs and RhoGAPs, RhoGDIs regulate Rho GTPase signaling. The flexible N-terminal domain of RhoGDIs interacts with the switch I and switch II domains of Rho GTPases, thus preventing the dissociation of GDP and interaction with RhoGEFs, RhoGAPs, and effector proteins. Therefore, RhoGDIs are considered to be negative regulators of Rho GTPases [10]. The C-terminus of RhoGDI forms a hydrophobic pocket, through which it binds to prenylated Rho GTPase. When cells are activated by various stimuli, Rho GTPases are released from RhoGDIs and integrate into the membrane, where they can be activated by RhoGEFs. Re-association between Rho GTPases and RhoGDIs facilitates the extraction of Rho GTPases from the membrane. Recent study demonstrates that RhoGDI can extract both GDP-bound and GTP-bound Rho GTPases, and that extraction of active Rho GTPases contributes to their spatial regulation [11]. RhoGDIs thus contribute to the recycling of Rho GTPases into the cytosol and other proper action sites on specific membrane compartments. When not bound to RhoGDIs, Rho GTPases are unstable, and rapidly degraded in the cytosol. Further, RhoGDIs protect Rho GTPases from proteasomal-degradation by preventing their interaction with ubiquitin ligases [12]. The interaction between RhoGDIs and Rho GTPases can be regulated by PTMs, such as phosphorylation and sumoylation [13]. Therefore, RhoGDIs are multifaceted modulators, regulating the spatiotemporal activity and stability of Rho GTPases through dynamic interactions.

For most Rho GTPases, including the Rho, Rac, and Cdc42 subfamilies, GDP/GTP cycling is regulated by RhoGEFs and RhoGAPs. However, there are atypical Rho family members that are activated via another mechanism. For example, the Rnd (Rnd1–3) and RhoBTB (RhoBTB1–3) subfamilies have amino acid substitutions in their GTPase domain that are not recognized by RhoGAPs, and thus, their GTP hydrolysis rate cannot be accelerated by these proteins. These subfamily members are thus constitutively active and GTP-bound, although they have intact GDP/GTP nucleotide exchange activity [14,15]. Similarly, RhoH has defective GTPase activity and is unable to hydrolyze GTP, and therefore is presumed to be constitutively active [16]. The RhoU/RhoV and RhoD/RhoF subfamilies represent another type of atypical Rho GTPases. These family members have high intrinsic guanine nucleotide exchange activity, and are assumed to be mostly GTP-bound in cells [17,18]. Because these atypical Rho GTPases also have GTPase activity and can cycle between GDP-bound and GTP-bound forms, they are considered to be the fast-cycling Rho GTPases [6,19,20].

## 3. Regulation of Rho GTPases by PTMs

Rho GTPases are regulated by various PTMs, including lipid modifications, phosphorylation, ubiquitination, and sumoylation (Figure 2). These modifications modulate their activity and protein levels, and are linked to anomalous Rho GTPase signaling in human cancers (Table 1).

### 3.1. Regulation of Rho GTPases by Lipid Modifications

Rho GTPases undergo two types of post-translational lipid modification, i.e., prenylation and palmitoylation, which regulate their trafficking and activity. Most Rho GTPases have the CAAX motif in their C-terminus. The AAX residues are removed by proteolysis, and the cysteine residue is then prenylated by farnesyltransferase or geranylgeranyltransferase, which attach a farnesyl or geranylgeranyl isoprenoid lipid moiety [41,42]. RhoGDIs can bind to the prenylated Rho GTPases through their hydrophobic pocket, and regulate their membrane targeting and signaling. Prenylation is therefore critical for the translocation of Rho GTPases into the membrane compartment and their biological activities [43,44]. Palmitoylation is another lipid modification that involves the covalent attachment of a fatty acid, such as palmitic acid, to a cysteine residue of Rho GTPase [45]. Instead of the CAAX motif, the atypical Rho GTPases RhoU and RhoV have a C-terminal CFV motif that can be palmitoylated, allowing them to transiently associate with membrane compartments [46]. RhoU and RhoV do not bind to RhoGDIs, because they are not prenylated. Dynamic and reversible palmitoylation might regulate their membrane translocation and activities [47].

### 3.2. Regulation of Rho GTPases by Phosphorylation

Numerous studies have been reported that Rho GTPase functions can be modulated by protein phosphorylation. Phosphorylation of RhoA on Ser188 by protein kinase A increases its binding affinity for RhoGDIs, leading to the release of RhoA from the plasma membrane and consequently, inhibition of RhoA activity [21,22]. cGMP-dependent protein kinase G and AMP kinase α1 also can phosphorylate RhoA Ser188, resulting in RhoA inactivation [23]. While the phosphorylation of RhoA Ser188 inhibits its activity thorough binding to RhoGDIs, this interaction in the cytosol also protects RhoA from ubiquitin-mediated proteasomal degradation [24]. A recent study reported that c-Met can phosphorylate RhoA on Tyr42 to promote RhoA ubiquitination and degradation [25]. The Ser71 and Tyr64 residues reside in the switch II region of Rac1, which is the binding site for GTP and regulatory proteins, such as RhoGEFs and RhoGDIs. AKT serine/threonine kinase phosphorylates Rac1 on Ser71 to decrease its GTP-binding activity [28]. Src and focal adhesion kinase can phosphorylate Tyr64 on Rac1, leading to the interaction of Rac1 with RhoGDIs and inhibition of its activity [27]. On the other hand, phosphorylation of Cdc42 on Tyr64 increases its association with RhoGDI1, which is necessary for cellular transformation [29]. Interestingly, Rac1 is phosphorylated on Thr108 by ERK, leading to its translocation to the nucleus [26]. Src-mediated Tyr254 phosphorylation on RhoU induces its translocation from the plasma membrane to the endosomal compartments and suppresses its biological activities [48]. Phosphorylation of Ser218 and Ser210 of Rnd3 by Rho-associated coiled coil-containing protein kinase (ROCK) and protein kinase C, respectively, promotes the interaction of Rnd3 with 14-3-3, which increases its stability and leads to its translocation to the cytosol [49,50,51].

### 3.3. Regulation of Rho GTPases by Ubiquitination and Sumoylation

Ubiquitination is the covalent attachment of ubiquitin, leading to degradation by the ubiquitin-proteasome system, which regulates a broad range of cellular functions, including cell migration, proliferation, and survival. Several Rho GTPases, such as RhoA, Rac1, and Cdc42, are subjected to ubiquitination. In addition, this modification regulates the spatiotemporal activities of Rho GTPases [52,53]. RhoA is ubiquitylated by several E3 ubiquitin ligase complexes, including SMAD ubiquitination regulatory factor1 (Smurf1), SKP1-CUL1-F-box (SCF)^FBXL19^ and CUL3/BACURD [31,32,54,55]. Smurf1, a HECT domain Ub ligase, catalyzes the ubiquitination of active GTP-RhoA, leading to its degradation at the specific cellular protrusion of migrating cells. CUL/BACURD complexes catalyze the ubiquitination of GDP-RhoA, whereas SCF^FBXL19^ targets both GDP-RhoA and GTP-RhoA for ubiquitination and degradation. CUL3 directly binds to RhoBTB2 and RhoBTB3 and mediates their ubiquitination [33,56]. Rac1 can also undergo ubiquitination by several E3 ubiquitin complexes: Cullin-1, the HECT domain, and ankyrin repeat containing E3 ubiquitin ligase1 (HACE1), X-linked inhibitor of apoptosis and cellular IAP1, and SCF^FBXL19^ complexes [34,35,36,57,58,59]. HACE1 specifically catalyzes the ubiquitination of active Rac1, resulting in its proteasomal degradation.

Sumoylation is a post-translational modification that involves the addition Small ubiquitin-like modifier (SUMO), which affects protein localization, interaction, stability, and activity [60]. Rac1 can be post-transcriptionally modified by sumoylation. Protein inhibitor of STAT3 (PIAS3), a SUMO E3 ligase, catalyzes the sumoylation of Rac1 in response to hepatocyte growth factor, resulting in increased Rac1 activity and optimal cell migration [39]. As PIAS3 interacts more with GTP-Rac1 than with GDP-Rac1, sumoylation appears not to be essential for Rac1 activation, but rather affects the maintenance of the activated state [40,61].

## 4. Dysregulation of Rho GTPase Signaling in Human Cancers

Rho GTPases control a number of signaling pathways that regulate gene expression, cell migration, cell proliferation, and cell survival. Given their participation in various cellular functions, it is not surprising that they are associated with nearly all stages of cancer development and progression, including the dysregulation of cell proliferation, tissue invasion, angiogenesis, metastasis, and resistance to apoptosis [1,5]. Altered expression, mutations, and aberrant signaling of Rho GTPases are commonly observed in various cancer types, making them attractive targets for cancer therapy.

### 4.1. Altered Expression and Activity of Rho GTPases in Human Cancers

Upregulation of several Rho GTPase family members with oncogenic activity and downregulation of other members with tumor-suppressor activity are frequently observed in human cancers, and are involved in cellular transformation and malignancy [62]. While most Rho GTPases are aberrantly overexpressed in various cancers, the downregulation of certain Rho GTPases is also involved in cancer development and progression.

RhoA activity is deregulated in various human cancers, and seems to be involved in almost all stages of cancer progression. RhoA overexpression in breast cancer is correlated with proliferation, invasion, and angiogenesis. The knockdown of RhoA by siRNA inhibits the growth and angiogenesis of xenografted MDA-MB-231 breast cancer cell lines [63,64]. Moreover, RhoA is hyperactivated in gastric cancer tissues and cell lines, and is crucial for the cell cycle G1-S transition. The knockdown of RhoA effector protein mammalian diaphanous 1 (mDia1) by siRNA leads to an increases in expression of cell cycle inhibitor p21^Waf1/Cip1^ and p27^Kip1^, while the depletion of Rho-kinase (ROCK) reduced the expression and activities of CDK4 and CDK6 [65,66]. In recent study, RhoA and RhoC expression are elevated, while RhoB expression is downregulated or absent in gastric cancer tissues, compared to normal gastric tissues. Ectopic expression of RhoA promotes proliferation and RhoC overexpression enhances motility and invasiveness of SGC7901 and AGS human gastric cancer cell lines, while RhoB overexpression suppresses these malignant phenotypes [67,68]. RhoB expression is also reduced or absent in lung cancer, compared to normal lung tissues. RhoB overexpression in A549 human lung cancer cell lines suppress cell proliferation in vitro, and xenograft tumor growth in nude mice [69]. In contrast, RhoB is overexpressed in breast cancer and promote cell proliferation, migration, and angiogenesis [70]. Thus, RhoA and RhoC are generally believed to act as tumor promoters, whereas the role of RhoB in human cancer is somewhat controversial.

The overexpression and hyperactivation of Rac has been found in several human cancers [71]. For example, Rac1 is overexpressed in gastric cancer tissues and is correlated with differentiation, local invasion, and lymph node metastasis [72]. In a recent study, high Rac1 activity in gastric adenocarcinoma tissues is correlated with worse overall survival. The inhibition of Rac1 by shRNA, or NSC23766, reverses chemotherapy resistance in gastric adenocarcinoma cell lines [73]. Moreover, the knockdown of Rac1 by siRNA in MCF-7 and MDA-MB-468 human breast cancer cell lines reduces EGF-induced cell migration and invasion [74]. The depletion of Rac1 in Kras-induced pancreatic ductal adenocarcinoma in mice reduces tumor development and prolongs survival [75]. Rac1b, the constitutively active splice variant of Rac1, is highly expressed in colon, lung, and breast cancers, and plays a critical role in different stages of cancer progression [76,77]. Cdc42 expression is also upregulated in many human cancers [78]. For example, Cdc42 overexpression is found in colorectal cancer, and is associated with the potential tumor suppressor gene ID4 [79]. In addition to modulation of the transforming activities of oncogene, Cdc42 may also function as tumor suppressor in several human cancers. Loss of Cdc42 in liver leads to a chronic liver damage and development of hepatocellular carcinoma [80]. Furthermore, Rac1 and Cdc42 activities decrease in human pheochromocytomas, compared with the adjacent normal tissues, and correlates with FARP1 and ARHGEF1 expression [81]. The atypical Rho GTPases, RhoBTB1 and RhoBTB2, are downregulated in head and neck cancer and breast cancer, respectively, and have thus been described as tumor suppressors [82,83]. The downregulation of Rnd3 expression is observed in hepatocellular carcinoma, and is correlated with cancer progression and poor prognosis [84]. However, Rnd3 expression is upregulated in gastric cancer cells, leading to the promotion of epithelial-to-mesenchymal transition and multidrug resistance [85,86]. Therefore, the roles of individual Rho GTPases in human cancers are complex and are dependent on extracellular stimuli and signaling pathways in different cancer cell types or tumor stages.

### 4.2. Modulation of Rho GTPase Activity by Regulatory Proteins

As mentioned earlier, Rho GTPase activity is tightly regulated by three types of regulator proteins: RhoGEFs, RhoGAPs, and RhoGDIs. Numerus studies have suggested that altered expression or activity levels of these regulators are associated with cancer malignant phenotypes. Altered activity and expression of RhoGEFs are seen in human cancers, and cause aberrant activation of Rho GTPases. RhoGEFs, such as PDZ-RhoGEF, LARG, and p115GEF, are highly activated by aberrant signaling from G protein-coupled receptors or tyrosine kinase receptors, resulting in enhanced Rho GTPase activation [9]. Furthermore, overexpression of RhoGEFs, such as epithelial cell transforming sequence2 (Ect2), T-cell lymphoma invasion and metastasis (Tiam1), and Vav1-3, has been observed in numerous human cancers [87]. For example, Ect2 is highly expressed in lung, esophageal, and ovarian cancers, and is involved in tumor cell growth [88,89]. Ect2 expression is also upregulated in hepatocellular carcinoma and correlates with early tumor recurrence and patient survival. Knockdown of Ect2 attenuates ERK signaling via suppressing RhoA, activities, resulting in reduction of cell migration [90]. Tiam1 expression is upregulated in several cancers, including breast cancer and esophageal squamous cell carcinoma, and is correlated with poor prognosis and metastasis [91,92]. Tiam1 promotes Rac1 activation and cytoskeletal changes required for metastatic breast tumor cell invasion and migration [93].

In contrast to RhoGEFs, certain RhoGAPs, including those deleted in liver cancer (DLC-1) and p190RhoGAP, are reportedly generally downregulated in various cancers. Reduced DLC-1 expression through genomic deletion or epigenetic silencing is often observed in hepatocellular and cervical carcinoma [94,95,96]. The knockdown of DLC-1 increases GTP-bound RhoA and tumorigenic growth, and expression of constitutively active RhoA in these cells accelerate liver tumor formation, suggesting that DLC-1 acts as tumor suppressor [94]. ARHGAP5 (p190RhoGAP) is frequently deleted in oligodendrogliomas, and its overexpression inhibits glial proliferation and tumor formation by downregulating RhoA activity [97]. ARHGAP10 expression is downregulated in several human cancers [98,99,100,101]. The downregulation of ARHGAP10 is correlated with an advanced stage of breast cancer, and tumorigenicity of ovarian cancer cells [98,101]. Most RhoGAP proteins appear to function as tumor suppressors by regulating Rho GTPase activity. However, a few RhoGAPs, such as ARHGAP5, ARHGAP8, and ARHGAP42, are overexpressed in human cancers, and promote cell migration and invasion through undefined mechanisms [102,103,104]. Compared to the extensive knowledge of RhoGEFs, little is known about the detailed molecular mechanisms of most RhoGAPs. Therefore, it is necessary to study the specific role of each RhoGAP in different aspects of Rho GTPase function.

While the elevated expression of RhoGEFs and the reduced expression of RhoGAPs are commonly observed in human cancers, RhoGDI expression can be up- or downregulated in different cancer types, and their functions in cancer progression are relatively complex and controversial. RhoGDI1 expression is upregulated in hepatocellular carcinoma cell lines and tissues with highly metastatic potential, whereas it is downregulated and negatively correlated with malignant phenotypes in lung cancers [105,106]. RhoGDI1 has been found to be up- or downregulated in breast cancer in different studies [63,107,108], and RhoGDI2 expression is also differentially regulated in different human cancers. The downregulation of RhoGDI2 in bladder cancer is correlated with increased metastatic potential and poor prognosis [109]. However, several studies have indicated that RhoGDI2 expression is upregulated and correlated with enhanced Rac1 activity and malignant progression in gastric and ovarian cancers [110,111]. These contradictory roles of RhoGDIs are probably due to diversities in the cellular compartments they interact with in different cancer types and tumor stages.

### 4.3. Mutations of Rho GTPases in Human Cancers

Activating mutations of Ras GTPases have been frequently detected in human cancers [6], whereas Rho GTPase mutations were rarely found until a few years ago. However, high-throughput sequencing technology has led to the discovery of both gain-of-function and loss-of-function mutations of Rho GTPases in various human cancers. For example, substitution of proline 29 to serine (P29S) in Rac1 was observed in 4% to 9% of sun-exposed melanomas [112,113]. P29S mutation in the highly conserved switch I domain of Rac1 is a fast-cycling mutation that decreases GTPase activity and increases interaction with downstream effector proteins, thereby promoting the proliferation and migration of melanoma [114]. The fast-cycling Rac1 N92I mutation has been identified in HT1080 cells. Both Rac1 P29S and N92I are able to transform mouse fibroblasts and MCF10A cells [115]. In addition, constitutively active Rac1 mutants, such as G12V/R, Q61R/K, P34R, were observed in advanced germ cell tumors [116]. On the other hand, RhoA G17V mutation was found in 53.3% (24/45) or 67% (22/35) of angioimmunoblastic T cell lymphomas, and 18% (8/44) of peripheral T-cell lymphomas in different studies [117,118,119]. Introduction of RhoA G17V promotes proliferation in Jurkat T cells. Interestingly, RhoA G17V appears to act as a dominant-negative mutant that inhibits endogenous RhoA function, not as an activating mutant. Although the mechanism of this mutation on tumorigenesis is unclear, since RhoA signaling is inversely correlated with Rac1 in some cancers, one possibility is that reduced RhoA activity may promote tumorigenesis through enhanced Rac1 signaling [120]. Moreover, recurrent RhoA R5Q, G17E, and Y42C mutations were identified in approximately 25% of diffuse-type gastric carcinomas [121]. Cdc42 G12D mutation has been also found in melanoma, but its function has not been evaluated [112]. Additional somatic mutations in Cdc42, Q61R, and P34Q, were recently reported in well-differentiated papillary mesothelioma [122]. Further analysis of these mutations is required to determine their functional relevance in tumorigenesis and cancer progression.

## 5. Therapeutic Targeting of Rho GTPase Signaling

As mentioned earlier, the dysregulation of Rho GTPase signaling through changes in their regulators, PTMs, and direct mutations is involved in the multistep processes of malignancy, making this signaling pathway an attractive target for cancer therapeutic intervention (Table 2). In what follows, we highlight several strategies targeting their membrane localization and interaction with RhoGEFs or effectors to interfere with dysregulated Rho GTPase signaling in cancer.

Most Rho GTPases are post-translationally modified by the addition of a lipid moiety in their C-terminus, and this modification is necessary for their ability to mediate tumorigenic events. Targeting the enzymes responsible for this modification, such as farnesyl transferase or geranylgeranyl transferase, would prevent proper intracellular localization of Rho GTPases, which is necessary for their function [123]. For example, statins are inhibitors of 3-hydroxy-3-methylglutaryl-coenzyme A reductase, which is involved in cholesterol biosynthesis. Statins inhibit the synthesis of isoprenoid intermediates, thereby preventing the isoprenylation of Rho GTPases, and thus leading to the inhibition of their functions [124,125]. GGTI-2418 is a peptidomimetic small-molecule inhibitor of geranylgeranyltransferase I [126]. This inhibitor has recently entered phase I clinical trials, and no dose-limiting toxicities were reported, but the study was terminated prior to dose expansion, based on a sponsor decision [127]. It has been suggested that the major targets sensitive to farnesyl transferase inhibitors are probably other proteins [128]. Therefore, despite their entry into clinical trials, the low selectivity of the different GTPases is considered a disadvantage of the isoprenylation inhibitors.

Rho GTPases are activated by specific RhoGEFs in response to distinct stimuli, and thus, disrupting their interaction represents an attractive therapeutic strategy. Rhosin inhibits the interaction of RhoA with multiple RhoGEFs, including LARG, Dbl, p115RhoGEF, and PDZ-RhoGEF, and suppresses the invasiveness of breast cancer cells [129]. Y16 has been developed through high-throughput screening and inhibits RGS-containing RhoGEFs [130]. It inhibits mammary sphere formation of breast cancer cells [131] NSC23766 is a specific inhibitor of a subset of Rac-specific GEFs, such as Tiam1 and Trio. It suppresses proliferation and migration of PC-3 prostate cancer cells through inhibiting Rac1 activation [132]. EHop-016 was identified during the optimization of NSC23766, and suppresses cancer cell migration through interfering with the binding of Rac1 to Vav2 [133]. Cdc42 activity specific inhibitor (CASIN) was identified in cell-based assays, and inhibits Cdc42 interaction with intersectin, leading to suppression of colorectal cancer malignant progression [134,135]. ZCL278 is another selective Cdc42 small-molecule inhibitor that blocks the binding of Cdc42 to intersectin, thereby suppressing actin-based motility and migration in metastatic prostate cancer cells [136,137].

Targeting of downstream effector proteins is another strategy to inhibit Rho GTPase function in promoting malignancy. Y27632 and fasudil are selective inhibitors of the RhoA effector ROCK [138,139,140,141]. Y27632 treatment decreases invasion and alters cell survival of melanoma cells in vitro, and injection of Y-27632 in tumor-bearing mice resulted in a reduction of melanoma tumor growth [142]. Fasudil inhibit tumor invasion and metastasis in HT1080 and MDA-MB-231 tumor model [143]. RKI-1447 and Wf-536 possesses more selective and potent ability to block ROCK than Y27632 or fasudil, and inhibits the in vitro invasion and in vivo metastasis of breast cancer and melanoma cells [144,145]. Although several ROCK inhibitors have been developed and shown anti-tumor activity on several cancers, AT13148 is the only ROCK inhibitor that has entered a clinical trial [146]. A phase I study in advanced solid tumors was initiated in 2012 and was recently finished, but the results have not yet been reported (NCT01585701). The PAK family, downstream effector of both Rac and Cdc42, is attractive target for drug discovery. FRAX597 was a potent inhibitor of PAKs identified by high-throughput screening and displays potent anti-tumor activity in vitro and in vivo [147]. Non-competitive inhibitor IPA-3 binds to the regulatory domain of group I PAKs, thereby promoting the inhibitory conformation [148]. Although IPA-3 suppresses tumor growth in hepatocellular carcinoma and prostate cancer xenograft model, it is unstable under physiological condition and thus unsuitable for further clinical development [149,150].

## 6. Conclusions and Future Perspectives

Rho GTPases regulate signaling pathways that are related to cancer development and progression. Although numerous studies have focused on identifying the modulators that regulate their activities, the detailed mechanisms in different cancer types remain unclear. In this review, we discussed how Rho GTPase activity and signaling are dysregulated in human cancers. Aberrant signaling of Rho GTPases is frequently observed in human cancers and has been attributed to a variety of mechanisms, such as overexpression of some of the members with oncogenic activity, downregulation of other members with tumor-suppressive activity, and changes in upstream regulators and effector proteins. PTMs, including lipid modification, phosphorylation, sumoylation, and ubiquitination, are highly dynamic processes that can regulate Rho GTPase signaling pathways related to cancer progression. Given the significant contribution of Rho GTPases in tumor malignancy and the recent occurrence of genetic mutations in these proteins, targeting these signaling molecules is considered a promising strategy for cancer treatment. Although several inhibitors have been developed, their clinical applicability is currently limited. Advancements in rational drug design with extensive understanding of Rho GTPases functions in pathological conditions will improve their effectiveness and target specificity for therapeutic intervention.

## Figures and Tables

**Figure 1 cancers-12-01179-f001:**
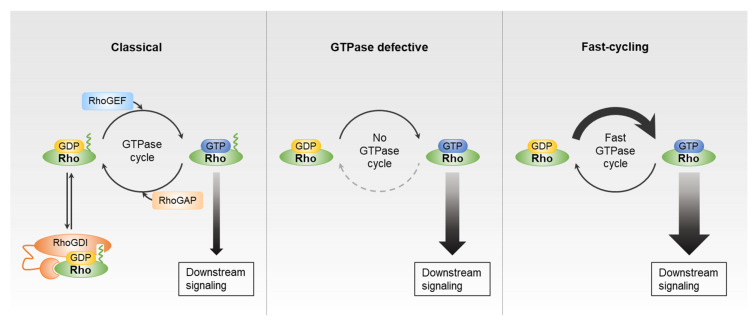
Guanosine diphosphate (GDP)-guanosine triphosphate (GTP) exchange cycle of Rho GTPases. Classic Rho GTPases, such as Rho, Rac, and Cdc42 subfamilies, follow GDP/GTP cycling regulated by Rho guanine nucleotide exchange factors (RhoGEFs), Rho GTPase-activating proteins (RhoGAPs), and Rho guanine nucleotide dissociation inhibitors (RhoGDIs). Rnd and RhoBTB subfamilies cannot be accelerated by RhoGAPs. The RhoU/RhoV and RhoD/RhoF subfamilies have high intrinsic GTP/GDP exchange activity.

**Figure 2 cancers-12-01179-f002:**
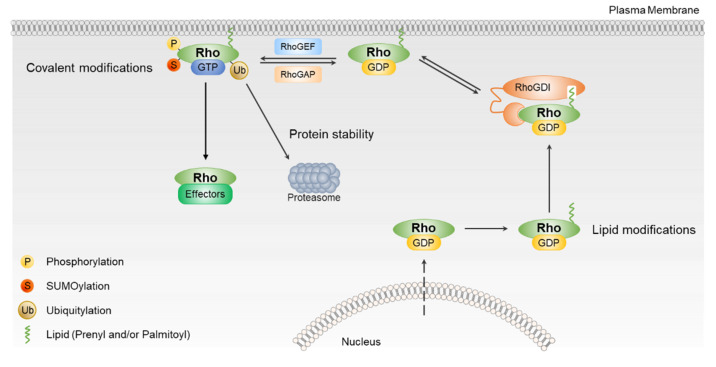
The regulation of Rho GTPases activation. Most Rho GTPases are regulated by RhoGEFs, RhoGAPs, and RhoGDIs. Rho GTPases are also regulated by post-translational modification, including phosphorylation, sumoylation, and ubiquitination. Post-translational modifications (PTMs) of Rho GTPases regulate their intracellular localization, stability, and ability to signal to downstream effector.

**Table 1 cancers-12-01179-t001:** Post-translational modifications of Rho GTPases.

PTMs	Rho GTPase	Regulator	Target Site	Effects	Refs
Phosphorylation	RhoA	PKA	S188	Increasing interaction with RhoGDI	[21,22]
				Decreases binding to ROCK effector	
				Protection of RhoA from degradation	
		PKG	S188	Translocation to the cytosol	[23]
				Protection of RhoA from degradation	
		PKC	T127 and S188	Translocation to the plasma membrane	[24]
				Protection of RhoA from degradation	
		c-Met	Tyr42	Proteasomal degradation	[25]
	Rac1	ERK	T108	RAC1 for translocation to the nucleus	[26]
		FAK	Y64	Inhibit RAC1 activity	[27]
		SRC	Y64	Inhibit RAC1 activity	[27]
		AKT	S71	Inhibits RAC1 activity	[28]
	Cdc42	SRC	Y64	Increasing interaction with RhoGDI	[29]
		PKA	S185	Increasing interaction with RhoGDI	[30]
Ubiquitination	RhoA	SMURF1	K6,K7 and K51	Proteasomal degradation	[31]
		SCF	K135	Proteasomal degradation in a ERK-dependent manner	[32]
		CUL3	ND	Proteasomal degradation of GDP-bound inactive RhoA	[33]
	Rac1	XIAP and clAP1	K147	Proteasomal degradation	[34]
		HACE1	K147	Targets GTP-bound, active RAC1 for degradation	[35]
		SCF	K166	Proteasomal degradation in a AKT-dependent manner	[36]
Transglutamination	RhoA	CNF1	E63	Constitutive activation	[37]
	Rac1	CNF1	E61	Constitutive activation	[37]
	Cdc42	CNF1	E61	Constitutive activation	[37]
AMPylation	RhoA	HYPE	T37	Suppress effector binding	[38]
	Rac1	HYPE	T35	Suppress effector binding	[38]
	Cdc42	HYPE	T35	Suppress effector binding	[38]
SUMOylation	Rac1	PIAS3	K183,K184,K186 and K188	Increased GTP binding and RAC1 activation	[39,40]

**Table 2 cancers-12-01179-t002:** Selected inhibitors of Rho GTPase signaling.

Drug Name	Target Protein	Mechanism	Refs
Rhosin	RhoA	Inhibit GEF biding	[129]
Y16	LARG	Inhibit RhoA binding	[130]
NSC23766	Rac1	Inhibit GEF binding	[132]
EHop-016	Rac1	Derivative of NSC23766	[133]
CASIN	Cdc42	Inhibit GEF binding	[134,135]
ZCL278	Cdc42	Inhibit GEF binding	[136,137]
Y-27632	ROCK	Compete with ATP	[140]
Wf-536	ROCK	Derivative of Y-27632	[144]
Fasudil	ROCK	Compete with ATP	[141]
RKI-1447	ROCK	Compete with ATP	[144]
AT13148	ROCK	Compete with ATP	[146]
K252	PAK	Compete with ATP	[151]
OSU-03012	PAK	Compete with ATP	[152]
PF-3758309	PAK	Compete with ATP	[153]
FRAX597	PAK	Compete with ATP	[147]
IPA-3	PAK	Promote auto-inhibited conformation	[148]

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
