# Peer review of "Dysregulation of Rho GTPases in Human Cancers"

_cancers, 2020, doi:10.3390/cancers12051179_

Round 1

Reviewer 1 Report

The review entitled « Dysregulation of Rho GTPases in human cancers » by Jung et al. aimed at summarizing recent findings regarding the role of Rho GTPases in cancer progression and the precise mechanisms controlling their activities. Although the review is well written, I have concerns about the 4th part of the review which address the dysregulation of Rho GTPases in cancers. I would also suggest emphasizing 5th part on targeting Rho GTPases which has been less addressed in the literature.

The 4th part does not provide enough experimental details and information to discriminate between conclusions inferred from in vitro or in vivo experiments. How Rho GTPases are related to the pathology is therefore unclear and information presented does not allow for a critical overview of Rho GTPase involvement in the cancer pathology. In addition, there is reference to outdated reviews (Fritz ey al., 1999; Gomez del Pulgar et al., 2005). More recent and extensive reviews are already available (Zandkivili et al., Oncogene 2016; Orgaz et al., Small GTPases 2014 and/or Haga, Ridley, 2016, Small GTPases; Svensmark, Brakebush, Oncogene, 2019) and would be more appropriate. The paragraph should be restructured to take into account those references. Like too many reviews on Rho GTPases in pathologies, some paragraph consists of a descriptive list of changes in expression levels of Rho GTPases in various cancers. Such a list can be summarized in a table. Text should focus on more convincing studies (where activities of Rho GTPases have been clearly measured for example) and present key data that involve Rho GTPases in tumor developments. The fact that Rho GTPases activities rather than its expression levels are critical to tumor progression/development should be emphasized and discussed. It has consequences on the way Rho GTPases dysregulation in cancers should be detected since at constant expression levels, Rho GTPases might be up or down regulated by GEF and GAPs (Croise et al., 2016, Endocrine-Related Cancer).

The 5th part needs to be developed. Some compounds are missing such as potential ROCK inhibitor (Wf-536) and Rho GTPases inhibitors (Azathioprine, Rocaglamide)

Specific comments

  • Line 72: “Most Rho GTPases are bound to RhoGDI” is confusing since evidences of Rho GTPases binding to RhoGDI concern to my knowledge only Rho, Rac and Cdc42. For this GTPases, the pool of Rho proteins represent indeed indeed 90%. This has to be clarified.
  • Line 95: The reference cited is not correct since it is a review on Rnd proteins that are not fast-cycling. To be changed.
  • Line 95-96: “As atypical atypical Rho GTPases have been shown to bind to any RhoGEFs and RhoGAPs” is not true. Although the function is unclear, RhoU has been shown to bind to CdGAP and ArhGAP30 (Naji et al., 2011)
  • Figure 1, line 102: a point to be removed
  • Line141, 142: “resulting in a decrease in GTP-binding” is not correct. We can only tell that phosphorylated RhoU is not recognized by the probe used. It can be a decrease in GTP binding (which is unlikely for a fast cycling GTPase in cellular context) or once phosphorylated an impairment of RhoU binding to its target.
  • Line183: paragraph 4.1 can be summarized in a table to focus on key studies and highlight relation between Rho activities and cancer developments
  • Line 190,191: outdated review 56 and 57 should be changed and paragraph should be also improved according to new references used.
  • Line 211: this paragraph lacks information about the consequence of regulator dysregulation on Rho GTPase activities in cancer. This is important information not to say critical since RhoGEF and RhoGAP are multimodular proteins with potential function independent on Rho activities.
  • Line 220;”founded” to be corrected
  • Line 302: “It the suppresses”: the to be removed
  • Line 310-317: Experimental context are not specified. Since this part focus on therapeutic targeting of Rho GTPase signaling, it is important to discriminate between results obtained from in vitro or in vivo assays.
  • Line 312: “beneficial effects have been extensively studied in various cancers” is overstated. In addition, cited ref 120 and 121 are not related to cancer studies.

Author Response

Apr 24, 2020

Assistant Editor of Cancers

Dear Dr. Ana Fenesan,

We are submitting a revised manuscript (Cancers-764510) entitled “Dysregulation of Rho GTPases in Human Cancers” for Cancers. We greatly appreciate the suggestions and comments of the reviewers. We have addressed their concerns and modified the manuscript accordingly. The modified manuscript was highlighted in red for ease of identification by the reviewers.

Specific comments and responses

Reviewer 1:

The review entitled « Dysregulation of Rho GTPases in human cancers by Jung et al. aimed at summarizing recent findings regarding the role of Rho GTPases in cancer progression and the precise mechanisms controlling their activities. Although the review is well written, I have concerns about the 4th part of the review which address the dysregulation of Rho GTPases in cancers. I would also suggest emphasizing 5th part on targeting Rho GTPases which has been less addressed in the literature.

The 4th part does not provide enough experimental details and information to discriminate between conclusions inferred from in vitro or in vivo experiments. How Rho GTPases are related to the pathology is therefore unclear and information presented does not allow for a critical overview of Rho GTPase involvement in the cancer pathology. In addition, there is reference to outdated reviews (Fritz ey al., 1999; Gomez del Pulgar et al., 2005). More recent and extensive reviews are already available (Zandkivili et al., Oncogene 2016; Orgaz et al., Small GTPases 2014 and/or Haga, Ridley, 2016, Small GTPases; Svensmark, Brakebush, Oncogene, 2019) and would be more appropriate. The paragraph should be restructured to take into account those references. Like too many reviews on Rho GTPases in pathologies, some paragraph consists of a descriptive list of changes in expression levels of Rho GTPases in various cancers. Such a list can be summarized in a table. Text should focus on more convincing studies (where activities of Rho GTPases have been clearly measured for example) and present key data that involve Rho GTPases in tumor developments. The fact that Rho GTPases activities rather than its expression levels are critical to tumor progression/development should be emphasized and discussed. It has consequences on the way Rho GTPases dysregulation in cancers should be detected since at constant expression levels, Rho GTPases might be up or down regulated by GEF and GAPs (Croise et al., 2016, Endocrine-Related Cancer).

The 5th part needs to be developed. Some compounds are missing such as potential ROCK inhibitor (Wf-536) and Rho GTPases inhibitors (Azathioprine, Rocaglamide)

Response: Thanks the reviewer for valuable suggestion. We have now restructured 4th part and 5th parts (line 205-245 and line 336-366). Here, we have added and discussed recent reference and missing inhibitors suggested by the reviewer 1.

Specific comments

1) Line 72: “Most Rho GTPases are bound to RhoGDI” is confusing since evidences of Rho GTPases binding to RhoGDI concern to my knowledge only Rho, Rac and Cdc42. For this GTPases, the pool of Rho proteins represent indeed indeed 90%. This has to be clarified.

Response: We have removed this sentence to eliminate reader’s confusion

2) Line 95: The reference cited is not correct since it is a review on Rnd proteins that are not fast-cycling. To be changed.

Response: We thank the reviewer for making us aware our error. We have changed it to correct reference. (line 95-96)

3) Line 95-96: “As atypical atypical Rho GTPases have been shown to bind to any RhoGEFs and RhoGAPs” is not true. Although the function is unclear, RhoU has been shown to bind to CdGAP and ArhGAP30 (Naji et al., 2011)

Response: We have now removed this sentence

4) Figure 1, line 102: a point to be removed

Response: We thank the reviewer for making us aware our error. We have removed it (line 100)

5) Line141, 142: “resulting in a decrease in GTP-binding” is not correct. We can only tell that phosphorylated RhoU is not recognized by the probe used. It can be a decrease in GTP binding (which is unlikely for a fast cycling GTPase in cellular context) or once phosphorylated an impairment of RhoU binding to its target.

Response: We have changed this sentence (line 139-141)

6) Line183: paragraph 4.1 can be summarized in a table to focus on key studies and highlight relation between Rho activities and cancer developments

Response: As reviewer suggested, we have restructured and highlighted relation between Rho GTPase activity and cancer progression in this paragraph (line 205-245)

7) Line 190,191: outdated review 56 and 57 should be changed and paragraph should be also improved according to new references used.

Response: According to reviewer’s suggestion, we restructured this paragraph with new references regarding RhoA functions in gastric cancer (line 207-216).

8) Line 211: this paragraph lacks information about the consequence of regulator dysregulation on Rho GTPase activities in cancer. This is important information not to say critical since RhoGEF and RhoGAP are multimodular proteins with potential function independent on Rho activities.

Response: The reviewer makes an excellent point. We now have addressed that RhoGEFs and RhoGAPs regulate cancer malignant phenotypes by regulating RhoGTPase activity (line 251-271).

9) Line 220;”founded” to be corrected

Response: We thank the reviewer for making us aware our error. We have now corrected the mistake and carefully checked if there was another mistake (line 256).

10) Line 302: “It the suppresses”: the to be removed

Response: We have removed “the” (line342).

11) Line 310-317: Experimental context are not specified. Since this part focus on therapeutic targeting of Rho GTPase signaling, it is important to discriminate between results obtained from in vitro or in vivo assays.

Response: According to reviewer’s suggestion, we specified experimental context (line 350-366)

12) Line 312: “beneficial effects have been extensively studied in various cancers” is overstated. In addition, cited ref 120 and 121 are not related to cancer studies.

Response: According to reviewer’s suggestion, we have now modified this sentence and changed to related references

Reviewer 2 Report

In this Review, Jung et al provide a summary and update on the dysregulation of Rho GTPases in cancer. This is a well-covered topic with several other recent reviews, but the information described in this ms provides a worthwhile perspective. The work is well organized and succinct, but there are a few issues that should be considered. 1) The description of Rho GDIs in section 2 is good to have, as these regulators are often left out of similar reviews. I would add, however, a reference to the recent paper by Golding et al in 2019 in eLife, as this work reinforces some much earlier observations that RhoGDIs might also interact with GTP-bound GTPases. 2) Line 88 is unclear. I would write “and thus, their GTP hydrolysis rate cannot be….” 3) Line 102/103 is unclear. I would write “have high intrinsic GTP/GDP exchange activity” 4) In section 3.2, I would add a reference to the phosphorylation of Rho Y42 by c-Met (Liu et al, J Path 2019). AS this event may destabilize RhoA, it could be an important feature to discuss. If space allows, it might also be useful to add a diagram depicting the post translational modifications. 5) Lines 310-317. I am surprised to see no mention of anti Pak agents. 5) In section 4, there is no mention of Rac1 G12V mutations, which as found in various germ cell tumors.

Author Response

In this Review, Jung et al provide a summary and update on the dysregulation of Rho GTPases in cancer. This is a well-covered topic with several other recent reviews, but the information described in this ms provides a worthwhile perspective. The work is well organized and succinct, but there are a few issues that should be considered.

1) The description of Rho GDIs in section 2 is good to have, as these regulators are often left out of similar reviews. I would add, however, a reference to the recent paper by Golding et al in 2019 in eLife, as this work reinforces some much earlier observations that RhoGDIs might also interact with GTP-bound GTPases.

Response: We have now added and discussed the great reference as suggested by the reviewer (line 75-77)

2) Line 88 is unclear. I would write “and thus, their GTP hydrolysis rate cannot be….”

Response: According to reviewer’s suggestion, we have changed this sentence (line 88).

3) Line 102/103 is unclear. I would write “have high intrinsic GTP/GDP exchange activity”

Response: We have changed this sentence as suggested by the reviewer (line 99-100)

4) In section 3.2, I would add a reference to the phosphorylation of Rho Y42 by c-Met (Liu et al, J Path 2019). AS this event may destabilize RhoA, it could be an important feature to discuss. If space allows, it might also be useful to add a diagram depicting the post translational modifications.

Response: We are grateful to the reviewer for these comments, and we have added and discussed the reference

5) Lines 310-317. I am surprised to see no mention of anti Pak agents.

Response: As reviewer suggested, we have more discussed PAK inhibitors in Section 5 (line 359-365).

6) In section 4, there is no mention of Rac1 G12V mutations, which as found in various germ cell tumors.

Response: We have now disscussed Rac1 G12V mutations as suggested by the reviewer (line 300-302)

Reviewer 3 Report

This is a timely review providing some overview about the state of the art on the role of RhoGTPase in tumour formation. However, I found that the reader is left alone with the description of numerous observations and that a core message for future strategies in the prevention of cancer becomes less clear.

The schematic outlines for the RhoGTPases GDP/GTP exchange cycles and their regulation is quite helpful but there are no supportive illustrations on the main topic, i.e. the dysregulation of RhoGTPase signaling in cancers and the therapeutic intervention.

I suggest that data on RhoGTPase signaling and pharmacological intervention in cancer, respectively should be assembled in one or two tables. Criteria in such tables could be the GTPase involved; Post translational mechanisms; Regulators: GEFs, GAP, GDIs; Tumour activation mechanism: e.g. tumour promoter, tumour suppressor; Cellular effects: proliferation, migration, survival; Tumour type; References

A second table could address drug targeting mechanisms containing Drug name, Target protein, Mechanism, Side effects/problems, References.

In the „Conclusion and future perspectives“ authors write in line 328„ ….These factors may co-operate in the regulation of the spatiotemporal activity of Rho GTPases in individual cells in response to different stimuli. Therefore, efforts to explore additional factors that modulate the signaling pathways of specific Rho GTPases will be helpful for understanding …………“ Could you be more specifically name „which“ additional factors you have in mind  or omit this sentence, otherwise.

Author Response

This is a timely review providing some overview about the state of the art on the role of RhoGTPase in tumour formation. However, I found that the reader is left alone with the description of numerous observations and that a core message for future strategies in the prevention of cancer becomes less clear.

The schematic outlines for the RhoGTPases GDP/GTP exchange cycles and their regulation is quite helpful but there are no supportive illustrations on the main topic, i.e. the dysregulation of RhoGTPase signaling in cancers and the therapeutic intervention. I suggest that data on RhoGTPase signaling and pharmacological intervention in cancer, respectively should be assembled in one or two tables. Criteria in such tables could be the GTPase involved; Post translational mechanisms; Regulators: GEFs, GAP, GDIs; Tumour activation mechanism: e.g. tumour promoter, tumour suppressor; Cellular effects: proliferation, migration, survival; Tumour type; References. A second table could address drug targeting mechanisms containing Drug name, Target protein, Mechanism, Side effects/problems, References.

Response: We appreciate the reviewer’s remarks. We have now added Table 1 (Post-translational modificationsof Rho GTPases on page 5) and Table 2 (Selected inhibitors of Rho GTPase signaling on page9) and reconstructed section 4.1 (altered expression and activity of Rho GTPases in human cancers, line 204-244).

In the „Conclusion and future perspectives“ authors write in line 328„ ….These factors may co-operate in the regulation of the spatiotemporal activity of Rho GTPases in individual cells in response to different stimuli. Therefore, efforts to explore additional factors that modulate the signaling pathways of specific Rho GTPases will be helpful for understanding …………“ Could you be more specifically name „which“ additional factors you have in mind or omit this sentence, otherwise.

Response: According to reviewer’s suggestion, we have deleted this sentence to avoid reader confusion.

Reviewer 4 Report

Haiyoung Jung et al. present a very synthetic review on the involvement of RhoGTPases in human cancers.
For a long time while the role of Ras GTPAses was well identified in tumor progression, the role of RhoGTPases remained poorly informed and little identified. This review comes at the right time as since 2012 many studies have reported a deregulation of RhoGTPases in cancer.
After a brief introduction, the review presents the direct regulatory proteins of RhoGTPases, the role of PTM, the modifications of RhoGTPases observed in cancer and finally the possible therapeutic interventions. The review is well-written, clear and quite exhaustive of the work in the field.

Minor criticisms:

- The title of section 2 may be somewhat confusing with the title of section3. This is because regulation by PTM also involves regulators.
I would suggest using the term seen later in the review of "regulator partners" and changing the title of section 2 to "Rho GTPases and their direct regulatory partners". This remains an intermediate solution because although I understand the authors' intentions in separating sections 2 and 3, regulation by PTM also involves regulatory partners. Knowing that proteasome degradations most often affect active forms and that most phosphorylations lead to inactivation of Rho GTPases, PTMs have more or less the same action as Rho GAPs. Knowing that from a biological point of view it is the set of reactions that conditions the activation time of Rho GTPases, I wonder if it is not more judicious to organize sections 2 and 3 into Rho GTPase activators and 3 into Rho GTPase inactivators.

-Journals are evaluated on the inclusion of the most recent articles in the field, but for some important points it is also useful to cite original articles rather than journal articles. Proteasome degradation is crucial and as important as regulation by GAPs and GEFs because it mainly affects the active form of Rho GTPases. Early work suggesting proteasome degradation of Rho GTPAses comes from studies involving the CNF-1 toxin that deamides Rho GTPases and transforms them into constitutively active forms (Olenik et al, J. Biol Chem.1999). Further work has shown an interaction of Rac1 with a ubiquitin ligase (Senadheera et al. Int J Mol Med. 2001) and a specific degradation of the active form of Rac1 by the proteasome (Kovacic et al. J Biol Chem. 2001). These initial steps in the discovery of the regulation of Rho GTPase by proteasomal degradation are worth mentioning.

-Information on downstream effector changes of Rho GTPases in cancer is missing. Indeed it is difficult to understand why the overexpression of RhoB acts as a tumor suppressor and the overexpression of RhoA and C acts as a tumor promoter. Does those Rho GTPases interact diffrently with ROCK and mDIA? A paragraph would be needed to explain which specific effectors are more mobilized during tumor progression and which effectors are more mobilized in an antitumor effect.

Author Response

Haiyoung Jung et al. present a very synthetic review on the involvement of RhoGTPases in human cancers. For a long time while the role of Ras GTPAses was well identified in tumor progression, the role of RhoGTPases remained poorly informed and little identified. This review comes at the right time as since 2012 many studies have reported a deregulation of RhoGTPases in cancer. After a brief introduction, the review presents the direct regulatory proteins of RhoGTPases, the role of PTM, the modifications of RhoGTPases observed in cancer and finally the possible therapeutic interventions. The review is well-written, clear and quite exhaustive of the work in the field.

Minor criticisms:

1) The title of section 2 may be somewhat confusing with the title of section3. This is because regulation by PTM also involves regulators. I would suggest using the term seen later in the review of "regulator partners" and changing the title of section 2 to "Rho GTPases and their direct regulatory partners". This remains an intermediate solution because although I understand the authors' intentions in separating sections 2 and 3, regulation by PTM also involves regulatory partners. Knowing that proteasome degradations most often affect active forms and that most phosphorylations lead to inactivation of Rho GTPases, PTMs have more or less the same action as Rho GAPs. Knowing that from a biological point of view it is the set of reactions that conditions the activation time of Rho GTPases, I wonder if it is not more judicious to organize sections 2 and 3 into Rho GTPase activators and 3 into Rho GTPase inactivators.

Response: We appreciate the reviewer’s remarks. We have now changed the title of section 2 to “Rho GTPases and their direct regulator partners" (line 52). Although we understand the reviewer's intention, we think it is good to separate sections 2 and 3 to highlight the regulation of Rho GTPase activity by PTMs. Instead we have not added a table 1 (post-translational modifications of Rho GTPases on page 5) to highlight these modifications.

2) Journals are evaluated on the inclusion of the most recent articles in the field, but for some important points it is also useful to cite original articles rather than journal articles. Proteasome degradation is crucial and as important as regulation by GAPs and GEFs because it mainly affects the active form of Rho GTPases. Early work suggesting proteasome degradation of Rho GTPAses comes from studies involving the CNF-1 toxin that deamides Rho GTPases and transforms them into constitutively active forms (Olenik et al, J. Biol Chem.1999). Further work has shown an interaction of Rac1 with a ubiquitin ligase (Senadheera et al. Int J Mol Med. 2001) and a specific degradation of the active form of Rac1 by the proteasome (Kovacic et al. J Biol Chem. 2001). These initial steps in the discovery of the regulation of Rho GTPase by proteasomal degradation are worth mentioning.

Response: We are grateful to the reviewer for these comments, and we have added and discussed the reference (line 156-158).

3) Information on downstream effector changes of Rho GTPases in cancer is missing. Indeed it is difficult to understand why the overexpression of RhoB acts as a tumor suppressor and the overexpression of RhoA and C acts as a tumor promoter. Does those Rho GTPases interact diffrently with ROCK and mDIA? A paragraph would be needed to explain which specific effectors are more mobilized during tumor progression and which effectors are more mobilized in an antitumor effect.

Response: According to reviewer’s suggestion, we have more discussed the role of effector proteins of Rho GTPases in human cancers (line 204-220)

We thank all the reviewers for their insightful and thorough comments and suggestions. We believe this reorganization would clarify the subject of this manuscript and improve readability. We look forward to a positive assessment of this revised manuscript.

Round 2

Reviewer 1 Report

Most of my concerns have been addressed.

Reviewer 2 Report

The authors have addressed my concerns.

Reviewer 3 Report

The authors have responded appropriately to the points I raised.